# A Deep Learning Approach for Gait Event Detection from a Single Shank-Worn IMU: Validation in Healthy and Neurological Cohorts

**DOI:** 10.3390/s22103859

**Published:** 2022-05-19

**Authors:** Robbin Romijnders, Elke Warmerdam, Clint Hansen, Gerhard Schmidt, Walter Maetzler

**Affiliations:** 1Department of Neurology, Kiel University, 24105 Kiel, Germany; r.romijnders@neurologie.uni-kiel.de (R.R.); c.hansen@neurologie.uni-kiel.de (C.H.); 2Innovative Implant Development (Fracture Healing), Division of Surgery, Saarland University, 66421 Homburg, Germany; elke.warmerdam@uni-saarland.de; 3Institute of Electrical Engineering and Information Technology, Faculty of Engineering, Kiel University, 24143 Kiel, Germany; gus@tf.uni-kiel.de

**Keywords:** gait, gait events, inertial measurement unit, deep learning

## Abstract

Many algorithms use 3D accelerometer and/or gyroscope data from inertial measurement unit (IMU) sensors to detect gait events (i.e., initial and final foot contact). However, these algorithms often require knowledge about sensor orientation and use empirically derived thresholds. As alignment cannot always be controlled for in ambulatory assessments, methods are needed that require little knowledge on sensor location and orientation, e.g., a convolutional neural network-based deep learning model. Therefore, 157 participants from healthy and neurologically diseased cohorts walked 5 m distances at slow, preferred, and fast walking speed, while data were collected from IMUs on the left and right ankle and shank. Gait events were detected and stride parameters were extracted using a deep learning model and an optoelectronic motion capture (OMC) system for reference. The deep learning model consisted of convolutional layers using dilated convolutions, followed by two independent fully connected layers to predict whether a time step corresponded to the event of initial contact (IC) or final contact (FC), respectively. Results showed a high detection rate for both initial and final contacts across sensor locations (recall ≥92%, precision ≥97%). Time agreement was excellent as witnessed from the median time error (0.005 s) and corresponding inter-quartile range (0.020 s). The extracted stride-specific parameters were in good agreement with parameters derived from the OMC system (maximum mean difference 0.003 s and corresponding maximum limits of agreement (−0.049 s, 0.051 s) for a 95% confidence level). Thus, the deep learning approach was considered a valid approach for detecting gait events and extracting stride-specific parameters with little knowledge on exact IMU location and orientation in conditions with and without walking pathologies due to neurological diseases.

## 1. Introduction

Gait deficits are common in older adults and possibly reflect the presence of an underlying neurodegenerative disease [1,2]. For example, conversion to Parkinson’s Disease [3] or conversion from mild cognitive impairment to Alzheimer’s Disease [4,5] are linked with changes in spatiotemporal gait parameters. Similarly, temporal gait parameters are different for stroke patients [6,7] and patients with multiple sclerosis [8,9] when compared to healthy controls. To objectively quantify gait deficits, stride-specific parameters such as stride time or stride length are often used [10]. The beginning and end of a stride are determined from two successive initial contacts (ICs) of the same foot [11,12]. The IC is when the foot contacts the ground and together with the instant at which the foot leaves the ground (final contact, FC), each stride can be divided in a stance and swing phase [13,14]. The events of IC and FC, also referred to as gait events, are commonly determined using force or pressure measuring devices [14] or marker-based optoelectronic motion capture systems (OMC; henceforth referred to as the marker-based system or method) [15,16]. These systems are relatively expensive and restricted to usage in expertise laboratories [17,18]. As there is increasing evidence that gait measured in the lab does not reflect daily-life gait [19,20,21], there is increasingly more interest in measurement systems that allow for continuous gait analysis in ambulatory settings. Therefore, the use of inertial measurement units (IMUs) is especially attractive, as these can be used to measure gait in ecologically valid environments, such as the home environment, thereby painting a more complete picture of health status [22,23] and providing clinical information that is complementary to standardized lab-based assessments [20,21,24,25].

Previous research suggests that gait event detection is more accurate using an IMU worn on a lower limb (e.g., shank or foot) compared to an IMU worn on the low back [26,27,28]. In order to get from abstract IMU sensor readings to clinically relevant gait parameters (i.e., from accelerations and angular velocities to stride times) [10], different algorithmic approaches have been developed in the last twenty years of clinical gait research. A recent study evaluated a cross-section of these algorithms for different sensor locations on the lower leg and foot [29]. The algorithms were categorized according to which signals were analyzed, for example, the angular velocity about the medio-lateral axis or the accelerations along vertical and antero-posterior axes. This means the sensor readings need to be linked with the anatomical axes, that is, one needs to know which sensor axis aligns with, for example, the medio-lateral axis. In most approaches, it is simply assumed that due to sensor attachment, the sensor axis aligns roughly with the anatomical axis of interest [30,31,32,33,34,35,36], or an additional calibration procedure (e.g., [37]) is required [29,38]. In ambulatory assessments, however, study participants often attach the sensor themselves, for example, after showering, and therefore the sensor location and alignment cannot be controlled for. Furthermore, it is unlikely that each time the sensor is (re-)attached, study participants, especially those with gait deficits, perform a calibration procedure that usually consists of holding a pre-defined pose and performing some known movement sequences [39,40].

Taken together, this drives the need for an approach that is invariant to sensor orientation and is applicable across a variety of pathological gait patterns. In the field of image analysis, similar requirements have been successfully addressed by algorithms that share a common underlying methodology referred to as deep learning (DL) [10,41,42], for example, in differentiating diseased from healthy cells [43]. The main advantage of DL is that rather than relying on expert-defined, hand-crafted features, the algorithm learns relevant data representations automatically [41,44]. Furthermore, DL approaches allow for individualization of the algorithm to a specific patient [45,46,47]. Previously, DL approaches for wearable IMU data have successfully been applied in classification of bradykinesia [44], detection of freezing of gait [48], and prediction of spatiotemporal gait parameters in people with osteoarthritis and total knee arthroplasty [49]. DL was used to detect gait events from marker-based motion capture and showed improved performance when compared to conventional, often heuristics-based, algorithms [50,51,52]. Another study used a DL approach to detect gait events from either three IMUs (worn on the low back, and both ankles) or a single IMU (worn on the low back) and showed that the time error was considerably smaller for the deep learning algorithm than for a commonly applied wavelet-based approach [53].

To the best of our knowledge, this is the first study that validates the performance of a DL approach for detecting gait events in a heterogeneous cohort of healthy and neurodegenerative diseases at multiple self-selected walking speeds from short walking distances using a single sensor setup that can be worn on either side, either laterally just above the ankle joint or proximally just below the knee joint.

The structure of the paper is as follows: in the Material and Methods section the data collection, data pre-processing, and the model architecture are introduced. The Results section presents the results from gait event detection and the subsequently extracted stride-specific gait parameters. In the Discussion, the results are set in relation to relevant literature, and finally in the Conclusions we outline the research content, results, and innovations.

## 2. Materials and Methods

### 2.1. Data Collection

Gait analyses were performed in the Universitätsklinikum Schleswig-Holstein (UKSH) campus Kiel, Germany. The study [54] was approved by the ethical committee of the medical faculty at the UKSH (no: D438/18). In total, data from 157 participants were included for the current analysis, including data from young adults (YA; age: 18–60 years), older adults (OA; age: >60 years), people with Parkinson’s Disease (PD; according to the UK Brain Bank criteria [55]), people with a recent (<4 weeks) symptomatic stroke (stroke), people with multiple sclerosis (MS; according to the McDonalds criteria [56]), people with chronic low back pain (cLBP), and people with diagnoses not fitting in any aforementioned groups or disorders with no explicit diagnosis (other) (Table 1). Inclusion criteria were an age of 18 years or older and the ability to walk independently without a walking aid. Participants were excluded from the study with a Montreal Cognitive Assessment [57] score < 15 and other movement disorders that affected mobility, as noticed by the clinical assessor.

Participants performed three walking trials consisting of walking 5 m at either (1) preferred speed (“*Please walk at your normal walking speed.*”), (2) slow speed (“*Please walk at half of your normal walking speed.*”), or (3) fast speed (“*Please walk as fast as possible, without running or falling.*”). The 5 m distance was marked with two cones on both ends, and participants were asked to start walking approximately two steps before the cones on one end, and stop walking approximately two steps after passing the cones on the other end.

For the current analysis, data from four IMUs (Noraxon USA Inc., myoMOTION, Scottsdale, AZ, USA) were considered, namely those that were attached laterally above the left and right ankle joint and those attached proximally at the left and right shank. IMUs were secured to participants using elastic bands with a special hold for the IMU. Furthermore, reflective markers were attached on top of the usual foot wear at the heel and toe of both feet (Figure 1). Marker data were recorded using a twelve-camera OMC system (Qualisys AB, Göteborg, Sweden) at a sampling frequency of 200 Hz. IMU data were recorded at the same sampling frequency, and both systems were synchronized using a TTL signal [54].

### 2.2. Data Pre-Processing

#### 2.2.1. Marker Data

First, data from both marker and IMU systems were cropped so that only data from within the 5 m distance were considered. Any gaps in the marker data were filled by interpolation making use of inter-correlations between markers [58,59]. The data were then low-pass filtered using a sixth-order Butterworth filter with a cut-off frequency of 20 Hz [60]. The filter was applied twice to the input data [61]. After filtering in the forward direction, the filtered sequence was reversed and run back through the filter [30]. The filtered data were differentiated to get velocity signals, and timings of ICs and FCs were determined from local maxima and minima in the heel and toe vertical velocity signals [62,63]. All identified ICs and FCs were manually checked using Qualisys Track Manager 2018.1 software (Qualisys AB, Göteborg, Sweden) and corrected if necessary [34,64]. The resulting annotated ICs and FCs were considered the true events (also labels or targets), and were used as reference timings to derive stride-specific gait parameters.

#### 2.2.2. IMU Data

The idea behind the deep learning approach was that a *model* was trained to predict the likelihood of an IC and FC, given accelerometer and gyroscope data from a single IMU. The data from a single sensor channel, e.g., the acceleration in forward direction, were denoted by $x_{d}={\left[\begin{array}{cccc} x_{d}\left[1\right] & x_{d}\left[2\right] & \cdots & x_{d}\left[N\right] \end{array}\right]}^{T}$, with *d* referring to the *d*th sensor channel (i.e., d=1,⋯,D) and *n* referring to the *n*th sample or time step (i.e., n=1,⋯,N). Similarly, the data from all *D* sensor channels at a given time instant *n*, were denoted by $x\left[n\right]={\left[\begin{array}{cccc} x_{1}\left[n\right] & x_{2}\left[n\right] & \cdots & x_{D}\left[n\right] \end{array}\right]}^{T}$. Data from all *D* channels, and for all *N* time steps, were then denoted by:(1)$$X=\left[\begin{array}{cccc} | & | & \; & |\\ x_{1} & x_{2} & \cdots & x_{D}\\ | & | & \; & | \end{array}\right]=\left[\begin{array}{cccc} x_{1}\left[1\right] & x_{2}\left[1\right] & \; & x_{D}\left[1\right]\\ x_{1}\left[2\right] & x_{2}\left[2\right] & \; & x_{D}\left[2\right]\\ \vdots & \vdots & \cdots & \vdots \\ x_{1}\left[N\right] & x_{2}\left[N\right] & \; & x_{D}\left[N\right] \end{array}\right],\;X\in R^{N\times D}$$

Likewise, the *labels* were denoted by:(2)$$y_{\mathrm{IC}}=\left[\begin{array}{c} y_{\mathrm{IC}}\left[1\right]\\ y_{\mathrm{IC}}\left[2\right]\\ \vdots \\ y_{\mathrm{IC}}\left[N\right] \end{array}\right],\;y_{\mathrm{FC}}=\left[\begin{array}{c} y_{\mathrm{FC}}\left[1\right]\\ y_{\mathrm{FC}}\left[2\right]\\ \vdots \\ y_{\mathrm{FC}}\left[N\right] \end{array}\right],\;y_{\mathrm{IC}/\mathrm{FC}}\left[n\right]\in \left[0,1\right]$$

The model was iteratively trained to learn a mapping $h_{\Theta }\left(X\right):X\to y$, where $h_{\Theta }$ was also referred to as the *hypothesis* parameterized by the weights, collectively denoted by Θ, and X was an array with raw sensor data from the 3-axis accelerometer and 3-axis gyroscope of a single sensor location.

All participant data were split into three independent datasets, namely a training set, a validation set, and a test set. Each set contained data from approximately one-third of the participants. Participants were randomly assigned to one of the sets, stratified by both group (i.e., diagnosis) and gender (Table 2). The training and validation set were used to train an optimal deep learning model. Test set data were not used for training the model or hyperparameter tuning. The results on the model’s performance were only based on the test set, and therefore reflected how good the model generalizes to new, *unseen* data.

Accelerometer and gyroscope data were normalized by subtracting the channel-wise mean and dividing by the channel-wise standard deviation. For the training and validation datasets, the data were then partitioned into equal length time windows [52] of 400 samples, with an overlap of 50% between successive windows (corresponding to 2 s windows, and 1 s overlap, respectively). For the test set, the complete trial was fed as input to the model for predicting ICs and FCs (hence the number of instances is the same as the number of trials, Table 2).

### 2.3. Model

#### 2.3.1. Model Architecture

The generic architecture for the deep learning model was based on a temporal convolutional network (TCN) [52,65,66]. The TCN consisted of a sequence of residual blocks with exponentially increasing dilation factor [66,67]. Each residual block was built from two sequences of a dilated convolutional layer [67], a batch normalization layer [68], a rectified linear unit (ReLU) activation layer, and a dropout layer [69] (Figure 2). The model was built in Python [70] using the high-level TensorFlow API Keras [65,71].

For the current analysis, only convolutions of the “same” type were considered [65], i.e., the model was non-causal and zero-padded to account for edge effects, and the likelihood of an IC or FC was based on input data both before and after the current sample, *n*:(3)$${\hat{y}}_{i}\left[n\right]=f\left(\cdots ,x[n-1],x[n],x[n+1],\cdots \right),\;i\in \left\{\mathrm{IC},\mathrm{FC}\right\}$$

The number of samples that the predictions at time *n* “sees” was referred to as the *receptive field* [72] and was a function of the kernel size and the dilation factors [65]. Dilation factors were always given as a sequence of increasing power of 2 [66,67,73].

The outputs of the TCN block were fed to two separate fully connected (FCN, or dense) layers, that were both followed by a sigmoid activation layer. Outputs were then predicted separately for ICs and FCs [52,53]. The mean squared error (MSE) was used as a loss function, and a gradient descent-based optimization algorithm with adaptive moment (Adam) optimizer was used to iteratively learn the weights [74,75].

#### 2.3.2. Hyperparameter Optimization

In order to find the best model architecture, hyperparameter tuning was perfomed using KerasTuner [76]. Here, the number of filters, the kernel size, and the maximum dilation factor (Table 3) were optimized for using a random search strategy [77].

The model architecture that resulted from the hyperparameter optimization (Table 4) was then trained on the combined set of training and validation data. The trained model was used to predict the likelihoods of ICs and FCs on the test set data.

### 2.4. Analysis

The predictions of the model on the test set data were compared with the labels from the test set. The model performance was evaluated for (1) overall detection performance, (2) time agreement between the predicted events and the (marker-based) annotated events, and (3) agreement between subsequently derived stride-specific gait parameters.

#### 2.4.1. Overall Detection Performance

The overall detection performance quantified how many of the annotated events were detected by the model (true positives, TP), how many of the annotated events were not detected (false negatives, FN), and how many events that were detected, were actually not annotated (false positives, FP). From these metrics, the recall (or sensitivity), precision, and F_1_ score were calculated as:(4)$$\mathrm{recall}=\frac{\mathrm{TP}}{\mathrm{TP}+\mathrm{FN}}$$
(5)$$\mathrm{precision}=\frac{\mathrm{TP}}{\mathrm{TP}+\mathrm{FP}}$$
(6)$$F_{1}\mathrm{score}=2\cdot \frac{\mathrm{recall}\cdot \mathrm{precision}}{\mathrm{recall}+\mathrm{precision}}$$

Here, recall represented the ratio of gait events that were detected, precision represented the ratio of detected gait events that were truly gait events, and F_1_ score was the harmonic mean of the recall and precision.

#### 2.4.2. Time Agreement

For all correctly detected gait events (TP, Section 2.4.1), time agreement was assessed by the time error between the annotated and detect gait event, which was defined as
(7)$$\mathrm{time}\mathrm{error}=t_{\mathrm{ref}}-t_{\mathrm{pred}}$$
with $t_{\mathrm{ref}}$ the gait event time from the marker-based annotations, and $t_{\mathrm{pred}}$ the gait event time from the model predictions. As a robust measure for the average time error and its spread, the median time error and the inter-quartile range (IQR) were reported [78].

#### 2.4.3. Stride-Specific Gait Parameters

For those trials for which all gait events were detected and no false positives were detected, the stride time, stance time, and swing were calculated. Stride was the time between two successive ICs of the same foot. Stance time was the time between a FC and the preceding IC of the same foot. Swing time was the time between the IC following the last FC of the same foot.

## 3. Results

### 3.1. Overall Detection Performance

The performance of detecting ICs and FCs was objectively quantified by the number of TPs, the number of FNs, and the number of FPs. From these numbers, recall, precision, and F_1_ score were calculated (Table 5).

For both ICs and FCs, recall is high for each of the sensor locations (i.e., ≥92%) and so is precision (i.e., ≥97%). Differences between the sensor locations are small, i.e., the minimum recall is 92% and the maximum recall is 97%, and the minimum precision is 97% and the maximum precision is 99%. The recall and precision result in F_1_ scores of ≥96% for ICs and ≥94% for FCs.

### 3.2. Time Agreement

Time agreement between the annotated and detected events was quantified for the TPs for each of the sensor locations (Figure 3). The median time error for each of the sensor locations and for both ICs and FCs was close to zero (Table 6), and the largest median time error was −0.005 s, corresponding to one sample period (at a sampling frequency of 200 Hz). The IQR was at most 0.020 s, corresponding to four sample periods.

### 3.3. Stride-Specific Gait Parameters

For those trials for which all gait events were correctly detected (and no false positives were detected), stride time, stance time, and swing time were calculated. The mean difference and the limits of agreement between the marker-based annotations and the model-based detections were calculated.

For all stride-specific gait parameters, and for all sensor locations, the mean difference was close to zero, i.e., the maximum mean difference was 0.003 s, namely for the calculated swing time of the right ankle (Table 7). Furthermore, for all gait parameters and for all sensor locations, the limits of agreement, based on a 95% confidence interval, were distributed around a zero-mean difference with the overall limits of agreement at −0.049 s and 0.051 s (Figure 4).

## 4. Discussion

The current study aimed to validate a deep learning approach for detecting gait events from a single IMU worn on the lower leg. Data from left and right ankle- and shank-worn IMUs were used for training a neural network to detect gait events from walking trials performed by healthy YA, healthy OA, participants diagnosed with PD, MS, or cLBP, participants who had a recent symptomatic stroke, and participants diagnosed with other neurological diseases. Participants walked a 5 m distance at three different self-selected walking speeds. The gait event timings that were predicted by the neural network were compared to a common reference method, i.e., OMC system, and clinically relevant stride-specific gait parameters were extracted.

A first measure for the model performance was given by the recall (how many annotated events were detected) and precision (how many detected events were annotated). For both ICs and FCs, a high recall (≥95%), high precision (≥98%), and high F_1_ score (≥94%) were observed, meaning that most events were detected and most detected events were actually true events. There was little difference in recall, precision, and F_1_ score between sensor locations (Table 5), confirming that the deep learning approach is relatively invariant to exact sensor localization. The values for recall, precision, and F_1_ score are comparable to recall (≥85%), precision (≥95%), F_1_ score (≥91%) from studies that detected gait events in OA, people with PD, and people with MS [34] or adults and hemiplegic patients [79].

Next, the time error, that is, the difference between the annotated event and the detected event, was of interest. For both ICs and FCs, and for all sensor locations, the observed time error was small, and the middle 50% of the time errors were within a range of $\left[-0.015,0.010\right]$ s (Table 6, Figure 3). These data showed that the deep learning-based approach is precise in detecting initial and final contacts. Time errors were slightly smaller than our previously reported results [34] that used a heuristics-based approach [30]. The heuristics-based approach determined ICs and FCs as local minima in the medio-lateral angular velocity [30], and it could be that these minima do not exactly coincide with the reference event timing as determined from the OMC systems. Previous studies that investigated the time error of IMU-based gait event detection reported a 95% confidence interval of [0.007, 0.013] s for IC, and [−0.005, 0.004] s for FC for young and older adults in treadmill and overground walking [80], or [−0.016, 0.001] s for IC or [0.037, 0.063] s for FC for typically developing children in overground walking [31]. Others reported the mean for the time error in healthy elderly subjects, subjects with PD, subjects with choreatic movement disorder, and hemiparetic subjects, and found a maximum mean error of 0.011 s at normal walking speed, and 0.022 s at faster speed [36]. Similarly, for healthy subjects, a mean error of 0.017 s for ICs and −0.016 s for FCs were reported, whereas for a single transfemoral amputee the mean error was 0.012 s for ICs and −0.024 s for FCs for the intact limb [33]. The median and IQR of the time errors of the current study were in the same range as previously found by [53], and time errors were smaller than previously reported time errors from a continuous wavelet-based approach [79]. Hence, our proposed deep learning approach resulted in time errors that are in the same range or better than those from previous approaches, while not being restricted to an exact sensor location (left or right ankle, or shank) and sensor alignment.

From the correctly detected gait events, stride-specific gait parameters were derived. These are probably of greatest clinical relevance, as changes in stride-specific gait parameters have been linked directly with disease onset and progression [3,4,5,6,7,8,9]. Therefore, stride time, stance time, and swing time were calculated, and the differences between the deep learning-based approach and the marker-based reference method were quantified (Table 7, Figure 4). The limits of agreement for a 95% confidence interval were calculated, and for all metrics the zero-mean difference was enclosed in the limit of agreement. These data provided evidence that the deep learning-based model was able to derive stride-specific gait parameters. The differences between the deep learning model-based stride parameters and the marker-based stride parameters were in a similar range as a recent study that compared IMU-derived stride parameters against stride parameters obtained with a pressure sensing walkway [29,53] and were also in the same range as results from a study that compared IMU-derived stride parameters with stride parameters obtained with an OMC systems [64]. The mean error was lower than the mean errors reported for stance and swing time (0.011 s and 0.011 s, respectively) across elderly subjects, subjects with PD, subjects with choreatic movement disorder, and hemiparetic subjects [36].

The main limitations of the current study were that only walking trials involving straight-line walking were considered, and the walking distance was relatively short. Therefore, it may be that the observed gait patterns from these walking trials are not fully representative of gait patterns observed in daily life [19,20,21]. However, as the deep learning-based approach does not rely on fixed thresholds or assumptions of which sensor axis is used, it is theoretically transferable and scalable to other conditions if input data and corresponding labels can be provided.

Furthermore, although the proposed approach allows for relatively arbitrary sensor placement on the lower leg, it was not investigated to what extent participants would be willing to wear such a sensor for a prolonged period of time in the home environment. Previous research found that user acceptance and adherence to wearing IMUs was generally high in people with neurodegenerative diseases [81,82,83,84], although reduced adherence was linked with multi-day wear [82] and wearing multiple sensors [85].

## 5. Conclusions

In this study we have validated a DL-based approach to detect gait events and subsequently extract clinically relevant stride-specific gait parameters from a single IMU worn either laterally above the ankle joint or proximal below the knee joint. Performance analysis showed an excellent detection rate and low time errors in both event detection and stride parameter calculation for different walking speeds and across both healthy and neurological cohorts. Compared to relevant approaches that detected gait events from an ankle- or shank-worn IMU, the DL approach reached a performance that was on par or better, and it did not rely on expert-defined, hand-crafted features or empirically derived thresholds. The performance of the DL approach was not affected by the exact sensor placement and orientation, and hence is less obtrusive for potential applications in long-term continuous monitoring. In contrast to previous approaches, it allows for personalization of the network to individual study participants and is easily transferable even to other sensor placement locations (e.g., a foot-worn or low back-worn IMU) without the need for rethinking the set of decision rules and thresholds. Our next step is to further develop and validate these methods with real-life walking sequences in patients with neurodegenerative diseases.

## Figures and Tables

**Figure 1 sensors-22-03859-f001:**
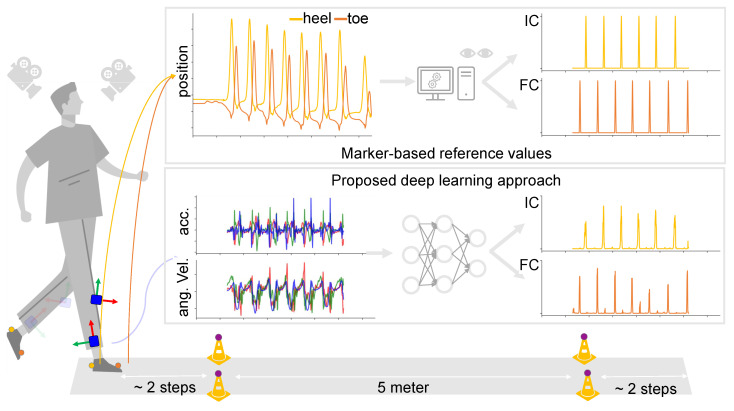
Schematic depiction (Picture from: https://www.vecteezy.com/free-vector/man-walking, accessed on 11 November 2021) of the current study. Study participants wore IMUs on the ankle and shanks, and reflective markers were adhered on the heel and toe of usual footwear (illustrated on the **left**). Marker data were used to obtain reference values for the timings of initial and final contacts (**top**), where accelerometer and gyroscope data from each tracked point were inputted to a neural network that predicted timings of the same initial and final contacts (**bottom**).

**Figure 2 sensors-22-03859-f002:**
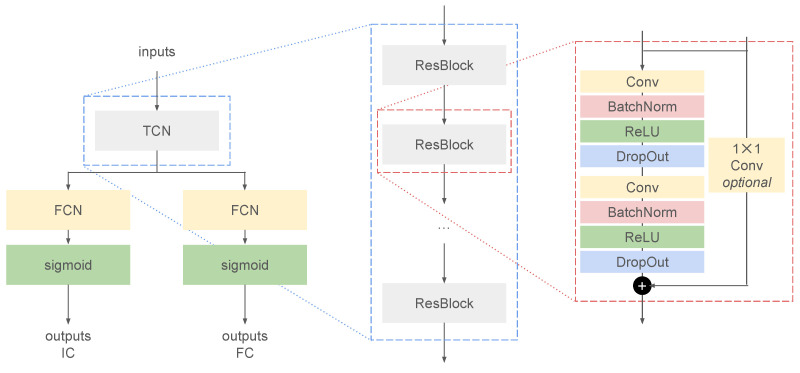
The generic model architecture of the deep learning model to predict initial contacts (ICs) and final contacts (FCs). The inputs are the accelerometer and gyroscope data from a single inertial measurement unit, which are fed to a temporal convolutional network (TCN) (**left**). The TCN consisted of repeating residual blocks (ResBlocks) with exponentially increasing dilation factor (middle). Each ResBlock was built from two sequences of a convolutional layer (Conv), batch normalization layer (BatchNorm), a rectified linear unit activation layer (ReLU), and a dropout layer (DropOut) (**right**).

**Figure 3 sensors-22-03859-f003:**
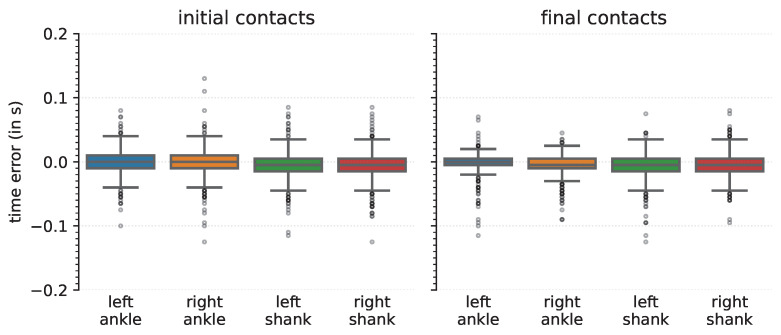
Time errors for initial (**left**) and final (**right**) contacts detection, for each of the different tracked points.

**Figure 4 sensors-22-03859-f004:**
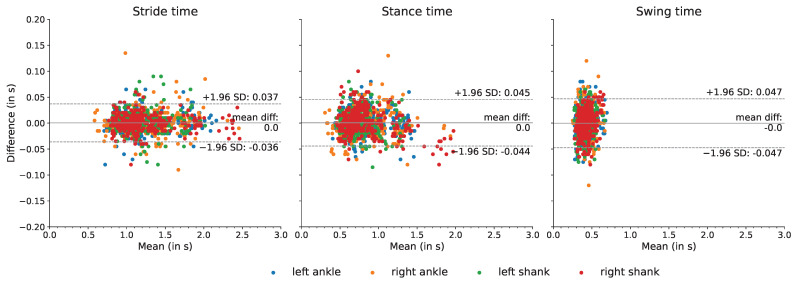
The agreement of extracted gait parameters between the sensor-based and marker-based methods. The differences between the stride-specific temporal gait parameters extracted from the marker-based and proposed approach are plotted against the means.

**Table 1 sensors-22-03859-t001:** Demographics data of the study participants. Age, height, and weight are presented as mean (standard deviation).

Group	Gender	Number of Participants	Age years	Height cm	Weight kg
YA	F	21	27 (7)	173 (5)	67 (9)
	M	21	29 (9)	185 (8)	80 (12)
OA	F	12	70 (6)	167 (6)	72 (17)
	M	10	73 (6)	180 (6)	83 (12)
PD	F	12	67 (6)	168 (7)	70 (15)
	M	19	61 (11)	178 (7)	86 (14)
MS	F	12	37 (10)	174 (9)	75 (9)
	M	9	42 (16)	189 (9)	96 (32)
stroke	F	4	66 (11)	160 (7)	65 (13)
	M	17	67 (18)	178 (7)	84 (15)
cLBP	F	3	64 (12)	166 (6)	65 (6)
	M	6	66 (17)	177 (8)	86 (14)
other	F	3	60 (16)	166 (4)	79 (19)
	M	8	68 (19)	182 (7)	85 (14)

**Group**: YA: younger adults, OA: older adults, PD: Parkinson’s Disease, MS: multiple sclerosis, cLBP: chronic low back pain; **Gender**: F: female, M: male.

**Table 2 sensors-22-03859-t002:** Overview of the total number of participants, walking trials, and number of instances in the training, validation, and test set. A detailed overview with exactly for which trial and sensor location valid data were available can be found at https://github.com/rmndrs89/my-gait-events-tcn, accessed on 1 April 2022.

Dataset	No. of Participants	No. of Trials	No. of Instances
Train	61	749	3366
Validation	48	564	2570
Test	48	620	620

**Table 3 sensors-22-03859-t003:** Model hyperparameters that were optimized for, and the corresponding sets of possible values.

Description	Possible Values
Number of filters	8, **16**, 32, 64, 128
Kernel size	3, **5**, 7
Dilations	[1, 2], **[1, 2, 4]**, [1, 2, 4, 8]

The hyperparameter values that were selected for the trained model to make predictions on the test set are shown in bold.

**Table 4 sensors-22-03859-t004:** Model layer hyperparameters.

Layer #	Layer Type	Hyperparameters	Output Shape
0	inputs		batch size × 400 × 6
1a	conv	no. of filters: 16	batch size × 400 × 16
		kernel size: 5	
		stride: 1	
		padding: same	
		dilation: 1	
1b	conv	no. of filters: 16	batch size × 400 × 16
		kernel size: 1	
		stride: 1	
		padding: same	
		dilation: 1	
2	conv	no. of filters: 16	batch size × 400 × 16
		kernel size: 5	
		stride: 1	
		padding: same	
		dilation: 1	
3	conv	no. of filters: 16	batch size × 400 × 16
		kernel size: 5	
		stride: 1	
		padding: same	
		dilation: 2	
4	conv	no. of filters: 16	batch size × 400 × 16
		kernel size: 5	
		stride: 1	
		padding: same	
		dilation: 2	
5	conv	no. of filters: 16	batch size × 400 × 16
		kernel size: 5	
		stride: 1	
		padding: same	
		dilation: 4	
6	conv	no. of filters: 16	batch size × 400 × 16
		kernel size: 5	
		stride: 1	
		padding: same	
		dilation: 4	
7a	dense	no. of units: 1	batch size × 400 × 1
7b	dense	no. of units: 1	batch size × 400 × 1

conv: convolutional layer.

**Table 5 sensors-22-03859-t005:** Overall detection performance for initial contacts and final contacts as quantified by recall, precision, and F_1_ score.

	Initial Contacts						Final Contacts					
Tracked Point	TP	FN	FP	Recall	Precision	F1	TP	FN	FP	Recall	Precision	F1
Left ankle	624	19	5	97%	99%	98%	606	32	10	95%	98%	97%
Right ankle	599	42	8	93%	99%	96%	614	17	12	97%	98%	98%
Left shank	605	38	15	94%	98%	96%	585	53	18	92%	97%	94%
Right shank	603	36	15	94%	98%	96%	595	30	9	95%	99%	97%

TP: true positives, FN: false negatives, FP: false positives, F1: F_1_ score.

**Table 6 sensors-22-03859-t006:** Time errors for the correctly detected gait events. Note that 0.005 s corresponds to 1 sample period, given the sampling frequency of 200 Hz.

	Initial Contacts		Final Contacts	
Tracked Point	Median s	IQR s	Median s	IQR s
Left ankle	0.000	0.020	0.000	0.010
Right ankle	0.000	0.020	−0.005	0.015
Left shank	−0.005	0.020	−0.005	0.020
Right shank	−0.003	0.020	−0.005	0.020

IQR: inter-quartile range.

**Table 7 sensors-22-03859-t007:** Time agreement between the stride-specific parameters.

Tracked Point	Parameters	Mean Difference s	Limits of Agreement (s, s)
Left ankle	stride time	0.001	(−0.035, 0.036)
	stance time	0.002	(−0.039, 0.042)
	swing time	−0.001	(−0.045, 0.043)
Right ankle	stride time	0.000	(−0.039, 0.040)
	stance time	−0.002	(−0.048, 0.044)
	swing time	0.003	(−0.046, 0.051)
Left shank	stride time	0.001	(−0.039, 0.041)
	stance time	0.002	(−0.043, 0.046)
	swing time	−0.001	(−0.049, 0.047)
Right shank	stride time	−0.000	(−0.031, 0.031)
	stance time	0.002	(−0.046, 0.049)
	swing time	−0.002	(−0.049, 0.046)

## Data Availability

Data from the first 10 participants are available online at https://github.com/neurogeriatricskiel/Validation-dataset (accessed on 11 November 2021). Additionally, we are preparing the open-source release of all data from the healthy younger and older adults. Data from patient groups can be shared upon reasonable request. The scripts are publicly available at the author’s personal GitHub, which can be found at https://github.com/rmndrs89/my-gait-events-tcn (accessed on 1 April 2022).

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
