# Peer review of "A Deep Learning Approach for Gait Event Detection from a Single Shank-Worn IMU: Validation in Healthy and Neurological Cohorts"

_sensors, 2022, doi:10.3390/s22103859_

Round 1

Reviewer 1 Report

The study is very interesting, with a clear clinical and social impact. The paper is very well-structured, very well-written and easy of follow and understand. The scientific methodology is sound and the results are relevant and interesting for both a clinical and a non-clinical audience. 

It would have been useful to have the patients' opinion on the acceptability of the device but this can be probably evaluated in a future study. If, however, such results are available, they can be added in the Results section or a discussion on this can be proposed in the discussion section.

Reviewer 2 Report

Overall, it is an interesting task. The task can be further improved by

  • In abstract section, need to mention the accuracy of algorithm
  • Need to improve abstract, more clearly mention the method.
  • In introduction section, cite some relevant article.
  • Iqbal, Muhammad Shahid, et al. "Deep learning recognition of diseased and normal cell representation." Transactions on Emerging Telecommunications Technologies(2020): e4017.

  • Add Background section, it will be easy to understand the article.
  • Figures 1, and 2 text is so small, not read able.
  • Clearly highlighted your proposed method and what new it has?
  • Tables, needed to be clearer and explain and also mention the accuracy equations.
  • Result and discussion needed to be more explaining and clear.
  • Method section is not clear, make it clearer and explain.
  • Add the discussion section and future problem, related to this topic
  • Add some more comparison with existing methods and show your method is better.  

This is an interesting topic and is generally well presented. The English expression, however, does need attention. I suggest that you get assistance with this and resubmit.

Reviewer 3 Report

 I have some comment as follow.

  1. In introduction, the authors should describe and refer what current deep learning methods are used for ‘Gait deficits’. What are the advantages and disadvantages of these methods. The author should give more research progress to illustrate the innovation and necessity of this research in introduction.
  2. The introduction describes the objectives but not the novelty. The authors should highlight the contributions in the introduction (not only in the conclusion).
  3. The structure of this paper should be briefly introduced in the ‘Introduction’.
  4. I would suggest that authors should use a summary table to describe for training set,validationand test set. It should be included the sample numbers, labels, sample size.etc
  5. What exactly is the input to the network? Is it the raw signals from the accelerometer and gyroscope? Are they input separately or as composite features after fusion? What is the size of the input signal? The author should explain in detail.
  6. Is the input signal  one-dimensional signal or image?
  7. The structureand parameters (stride, kenerl size, kenel number etc) of deep learning network should be described in detail in a table.
  8. The results of this method should be compared with existing methods to illustrate the advantages of proposed method.
  9. The conclusion should be improved to outline the research content, results and innovations.

Round 2

Reviewer 3 Report

The manuscript has been improved by previous comments. I suggest to publish.